# Enhancing Drought Tolerance and Fruit Characteristics in Tomato through Exogenous Melatonin Application

Qian Huang [1,2,3], Haijing Yan [1,2,3], Mingyuan You [1,2,3], Jinye Duan [1,2,3], Manling Chen [1,2,3], Yingjin Xing [1,2,3], Xiaohui Hu [1,2,3] and Xiaojing Li [1,2,3,*]

1   College of Horticulture, Northwest A&F University, Xianyang 712100, China;
    huang_qian1208@163.com (Q.H.); hjyan004@163.com (H.Y.); ymingyuan0510@163.com (M.Y.);
    zixin151877@163.com (J.D.); chenmanlingccc123@hotmail.com (M.C.); yingjin.xing@foxmail.com (Y.X.);
    hxh1977@163.com (X.H.)
2   Key Laboratory of Protected Horticultural Engineering in Northwest, Ministry of Agriculture,
    Xianyang 712100, China
3   Shaanxi Protected Agriculture Research Centre, Xianyang 712100, China
*   Correspondence: lixiaojing@nwafu.edu.cn

**Abstract:** Drought stress not only affects the growth and development of tomato seedlings but also leads to a significant decrease in tomato fruit yield. Previous studies have shown that melatonin plays a crucial role in regulating plant tolerance to drought stress. The present study was conducted to investigate the impact of exogenous melatonin on the growth and development of tomato seedlings under drought stress, as well as its potential in improving fruit yield and quality. Our findings demonstrate that drought stress strongly suppressed growth and biomass accumulation, reduced photosynthetic pigments, and inhibited photosynthesis. Conversely, melatonin treatment led to a notable increase in plant height, stem diameter, aboveground biomass, and relative water content of tomato seedlings by 16.67%, 7.39%, 10.58%, and 13.31%, respectively, compared to the drought treatment. Moreover, the chlorophyll content increased by 40.51%, and the net photosynthetic rate increased by 1.2 times. Furthermore, the application of melatonin under drought stress resulted in a decrease in osmoregulation substances, reduced accumulation of reactive oxygen species, and enhanced activity of antioxidant enzymes in tomato seedlings. Exogenous melatonin was also found to inhibit the expression of abscisic-acid-synthesis-related genes, resulting in a reduction in the abscisic acid content in tomato seedlings. Additionally, it significantly increased the root length, root surface area, and root vitality of the plants. When compared to drought treatment, tomato plants treated with melatonin exhibited a 61.92% increase in average yield and a 37.79% increase in fruit weight per plant. Furthermore, the organic acid content decreased by 23.77%, while soluble solids and sugars increased by 15.07% and 35.49%, respectively. These findings suggest that exogenous melatonin effectively alleviates the inhibition of photosynthesis and growth in tomato seedlings under drought stress. It achieves this by regulating the content of osmotic stress substances and the activity of antioxidant enzymes, thus enhancing the resistance of tomato seedlings to drought stress. Moreover, melatonin regulates root growth by mediating the biosynthesis of endogenous ABA, thereby improving the absorption and utilization efficiency of water and nutrients in plants. Consequently, it enhances tomato fruit yield and quality under drought stress.

**Keywords:** abscisic acid; melatonin; root morphology; tomato (*Solanum lycopersicum*); water stress; yield and quality

## 1. Introduction

In recent decades, drought has emerged as a major abiotic pressure that limits crop growth and development, making it a global issue that hampers agricultural productivity [1]. Drought stress can have various negative effects, including inhibiting cell division and enlargement [2], reducing photosynthesis [3], disrupting plant growth regulators balance [4],

and inducing oxidative damage [5]. These impacts can compromise cell membrane integrity and leaf water status and ultimately lead to decreased crop yield [6]. Additionally, drought alters stomatal activities and restricts nutrient uptake, further contributing to yield reduction [7]. The water deficit also triggers the generation of reactive oxygen species (ROS), which disrupts cellular redox regulation [8]. Excessive accumulation of ROS in plants under water-deficient conditions can cause significant damage to cellular organelles [9]. To enhance drought stress tolerance, plants employ antioxidant enzymes that effectively reduce ROS accumulation [10]. Furthermore, plants utilize osmotic adjustment as an important physiological mechanism to cope with water deficit [11]. They accumulate osmotic adjustment compounds, such as proline and soluble sugars, in order to scavenge reactive oxygen species (ROS) and prevent membrane peroxidation [12].

As a small molecular indole, melatonin (MT) is widely distributed in most plants and serves as a growth regulator with significant physiological functions. Research has shown that melatonin plays a role in enhancing plant growth and development, increasing agricultural yield, promoting seed germination, encouraging the development of lateral roots, delaying leaf aging, and influencing fruit ripeness [13,14]. Additionally, melatonin has been found to improve plant resilience to various abiotic conditions, such as drought, salinity, high temperature, low temperature, heavy metals, and oxidative stresses [15–18]. When plants are subjected to drought stress, melatonin helps increase stress resistance by maintaining water retention, reducing electrolyte leakage, and enhancing chlorophyll content and the net photosynthetic rate, thereby sustaining higher photosynthetic capability [19]. The external application of melatonin can enhance plant resilience to drought stress by regulating key antioxidant enzymes [20]. It has been observed that melatonin supplementation increases the activity of antioxidant enzymes in walnut leaves under drought stress, leading to improved scavenging of reactive oxygen species (ROS) and reduced oxidative damage [21]. Moreover, overexpression of SlCOMT1, which increases melatonin content, has been shown to promote stomatal closure and reduce water loss under drought stress by regulating abscisic acid (ABA) synthesis [22].

Tomato is a significant crop in both protected cultivation and field cultivation. However, drought stress poses a major limitation to plant productivity, affecting the growth and development of tomato. This stress leads to changes in its external morphology and physiological and biochemical indexes, ultimately impacting reproductive growth and yield quality [23]. Biostimulators, such as melatonin, have been found to positively influence plant growth regulation and resilience development by enhancing various physiological, biochemical, and molecular processes [24]. Melatonin, in particular, has shown promise in increasing plant drought stress tolerance [25]. Despite previous studies on the effects of melatonin on plant drought resistance, there is still a significant knowledge gap regarding the specific mechanisms by which melatonin enhances drought tolerance in tomato under water deficit conditions. Our study aimed to investigate various aspects, including plant state, photosynthetic performance, membrane damage of tomato seedlings, root architecture system, chloroplast structure, ROS homeostasis, and the expression of ABA-synthesis-related genes regulated by melatonin under drought stress. Furthermore, we examined the impact of exogenous melatonin on the growth and development of tomato seedlings under drought stress, as well as its potential to improve fruit yield and quality. By analyzing these factors, we aimed to uncover the pathways through which melatonin influences plant physiology and stress response.

## 2. Materials and Methods

### 2.1. Experimental Materials

Tomato materials were AC (*Solanum lycopersicum* L. cv. Ailsa Craig). We selected the full, uniform size of tomato seeds to clean (rinsing with water 3 times), disinfect (soaking in 10% sodium hypochlorite solution for 5 min), soak (soaking in water for 8–10 h), and germinate (the seeds after soaking were washed twice with water, wrapped in gauze or a towel, placed at 25–30 °C to promote budding, washed with water once a day, and,

after 2–3 days, small white buds can be sown). The seeds were seeded into the plug matrix and cultured in the greenhouse. When the seedlings reached the age of two main leaves and one cotyledon (about 15 days), tomato seedlings with healthy and consistent growth were selected and planted, and the planted seedlings were treated 3–4 days later. Melatonin was provided by Beijing Coolaber Technology Limited Company, purity > 99%, LOT: CM32112308.

### 2.2. Experimental Designs

All experiments were conducted at Northwest A&F University, Yangling, Shanxi Province, China (34°20′ N, 108°24′ E).

There were four groups in this experiment. The first group (CK) received normal watering, while the second group (T1) received normal watering with the addition of 100 μmol/L melatonin. The third group (T2) experienced drought conditions, and the fourth group (T3) underwent drought conditions with the addition of 100 μmol/L melatonin. Each group had five replicates. The treatments of T2 and T3 were treated with natural drought (no water was given after melatonin spraying until the plants wilted), while CK and T1 treatments were used to regulate the soil moisture. We weighed at 8:00 every morning and hydrated so that the weight of each nutrient bowl was about 160–170 g. Before the experiment, the measured field water capacity of the substrate was 122.58%, each nutrient bowl contained 80g of substrate, and the weight of the nutrient bowl was about 10–12 g. The weight of tomato seedlings was ignored, according to the total weight = nutrient bowl + substrate + water + tomato seedling, water = $(70 - 75\%) \times 122.58\% \times 80$ g. The experiment was carried out between two main leaves and three main leaves of tomato seedlings. Each treatment was treated with 10 mL of 100 μmol/L melatonin solution (dissolving 0.0116 g of melatonin in 500 mL of distilled water) or water continuously for three days. After the plants showed the stress phenotype (about 10 days), the samples were taken to determine the related indexes.

In the initial experiment, the effects of melatonin on tomato seedling drought tolerance were discovered. A field experiment was designed to further determine the effects of melatonin on tomato fruit yield and quality.

The seedlings were planted in a 23 cm × 24 cm pot with 2 kg substrate. The first melatonin treatment was carried out after transplanting the slow-growing tomato seedlings. First, 10 mL of melatonin solution or water was watered every day for 3 days. Throughout the growing period, the tomatoes were tended to in the field by watering with 500 mL of Hoagland nutrition solution every two weeks. By controlling the amount of irrigation, the soil water content of the control and T1 treatments was maintained at 70–75% of the fields' water holding capacity, while the soil water content of the drought treatments was 60–65%. Tomato seedlings were slow after transplantation, and the first melatonin treatment was started after the slow seedling was finished (each plant was administered 500 mL of melatonin solution with a concentration of 100 μmol/L, and the roots were irrigated continuously for 3 days). After normal field management, the second melatonin treatment was started during the fruit expansion stage (root irrigation every seven days; each plant was treated with 500 mL of 100 μmol/L melatonin solution). Other management measures were as usual. Five individuals with the same growth were selected for each treatment, flowering from June 10 to harvest at the end of August. Samples of each treatment type were rapidly frozen in liquid nitrogen and stored at −80 °C throughout the remainder of the experiment.

### 2.3. Measurement Items and Methods

2.3.1. Phenotypic Record of Tomato Seedlings and Determination of Related Agronomic Traits

The plant height and stem diameter were measured using a ruler and vernier caliper. After cleaning the tomato seedlings, the fresh weight (FW) was measured with a balance, and the dry weight was determined after drying. At the seedling growth stage, the relative

water content of the third leaf was determined [26]. We advanced the oven temperature to 100–105 °C. The freshly weighed leaves were placed in the oven together with the weighing dish and deoxidized at 100–105 °C for 10 min, and then the temperature of the oven was reduced to about 70–80 °C. Then, the leaves were baked to a constant weight. We removed the weighing dish with the blades, cooled to room temperature, and weighed. We subtracted the weight of the weighing dish from the weight after weighing, which is the dry weight (DW). The relative water content (%) = (FW − DW)/FW × 100.

### 2.3.2. Determination of Chlorophyll Content and Photosynthetic Index

The chlorophyll content was extracted by 80% acetone [27]. The net photosynthetic rate (Pn), transpiration rate (Tr), intercellular $CO_2$ concentration (Ci), and stomatal conductance (Gs) of well-grown fully expanded leaves under the tips of tomato plants were measured by portable photosynthetic meter (LI-COR Biosciences, Lincoln, NE, USA) from 9:00 to 11:00 on sunny days. The determination time intensity was set to 1000 $\mu mol \cdot m^{-2} \cdot s^{-1}$, the atmosphere temperature was 26 °C, and the atmosphere was 390 $\mu mol \cdot mol^{-1}$.

### 2.3.3. Determination of Physiological and Biochemical Indexes

The barbituric acid method was used to detect Malondialdehyde (MDA) [28]. The method for the determination of relative electrical conductivity of leaves referred to in [29] was used to rinse the leaves with distilled water at the last tip, and the filter paper absorbed the water. We took the blade with a hole punch in the symmetrical position on both sides of the vein, and put the circular blade into a plug calibration centrifugal tube with distilled water. The leaves were submerged in water for 8 mL and measured with a DDS-11A conductivity meter (METTLER TOLEDO, Switzerland) after 12 h. For electrical conductivity R1, the boiling water bath was heated for 20 min, and the conductivity R2 was measured after cooling. The relative conductivity (%) = R1/R2 × 100.

The Coomassie Brilliant Blue technique was used to quantify the content of soluble protein [30]. The anthrone colorimetric method was used to assess the content of soluble sugar present [31]. The acid ninhydrin method was used to determine the proline [32].

### 2.3.4. Determination of Hydrogen Peroxide and Superoxide Anion Content

We referred to the determination method of hydrogen peroxide ($H_2O_2$) [33]. The superoxide anion ($O_2^-$) content was measured using the hydroxylamine oxidation technique [34].

The histochemical detection of $H_2O_2$ was modified regarding the staining method [35]. After washing with distilled water, the fresh tomato leaves were immersed into 1 mg/mL 3,3′-diaminobenzidine tetrahydrochloride (DAB) in the dark for eight hours. After removing the dye solution, the leaves were put in a 3/1/1 mixture of ethanol, glacial acetic acid, and glycerin and boiled until all of the chlorophyll was removed before taking pictures.

The histochemical detection of $O_2^-$ was modified regarding the staining method [36]. After washing with distilled water, the fresh tomato leaves were immersed into 1 mg/mL nitrogen blue tetrazole (NBT) in the dark for six hours. After removing the dye solution, the leaves were put in a 3/1/1 mixture of ethanol, glacial acetic acid, and glycerin and boiled until all of the chlorophyll was removed before taking pictures.

### 2.3.5. Determination of Antioxidant Enzyme Activity

The activity of the Superoxide Dismutase (SOD), Peroxidase (POD), and Catalase (CAT) activity were assessed using a kit (Suzhou Keming Biotechnology Co., Ltd., Jiangsu, Suzhou, China).

### 2.3.6. Determination of Endogenous Hormones Content

The 0.1 g leaves were immediately frozen in liquid nitrogen and processed through a high-flux grinder to create a powder. A mixture of 1 mL isopropyl alcohol/water/hydrochloric acid (100/50/1) was used for extraction; after concentration,

1 mL methanol was added, and the membrane was shaken. The hormone content was determined by Liquid chromatography–mass spectrometer.

### 2.3.7. RNA Extraction and qRT-PCR

RNA was extracted from the leaves and roots of tomato seedlings with different treatments by Omega Plant RNA Kit. The RNA was reverse transcribed to cDNA, and its concentration and purity were subsequently determined for subsequent real-time fluorescent quantitative PCR.

A Taq SYBR Green qPCR Premix fluorescence quantitative kit was used. The reaction system was 20 mL: 10 μL 2× ChamQ Universal SYBR QPCR Premix, 1 μL cDNA (5-fold dilution), 0.4 μL upstream and downstream primers, and the rest was supplemented with water. The PCR reaction was carried out using a StepOnePlus Real-Time PCR apparatus, with the following conditions: 95 °C predenaturation for 0.5 min, followed by 40 cycles of 95 °C 10 s and 60 °C 30 s.

Actin was selected as the internal reference gene. The following table includes a list of the primer sequences (Table 1). The $2^{-\Delta\Delta CT}$ method was used to calculate the relative gene expression [37].

**Table 1.** Primer sequences used in the qRT-PCR test.

| Gene Name | Forward Primer (5′-3′) | Reverse Primer (5′-3′) |
| --- | --- | --- |
| Actin | GGGATGGAGAAGTTTGGTGGTGG | CTTCGACCAAGGGATGGTGTAGC |
| NCED1 | CAGAACTGCAAATTGTTAACGC | GGCGTTTATGAATGTTCCATGA |
| NCED2 | CGGAAAGATTAGTTCAAGAGCG | CACGGGCATAGAACAACAATAG |
| NCED3 | TCCACGACCCGAATAAAGTATC | GCATGCAAAAGCAATTAGGAAC |

### 2.3.8. Determination of Root Activity and Root Morphology

The root activity was determined through the Triphenyltetrazole chloride (TTC) method [38]. The roots were washed after 10 days of melatonin treatment, using a scanner (Espon Expression 1680 Scanner, Seiko Espon Corp., Tokyo, Japan), and they were scanned with the WinRHIZO roots Analysis system (Ver.2.0, Regent Instruments Inc., Quebec, QC, Canada) [39].

### 2.3.9. Determination of Fruit Quality Index

The content of organic acids was determined by NaOH titration [40]. The soluble solid content was determined using a portable solid state analyzer. The soluble sugar was determined through sulfuric acid-anthrone colorimetry [30]. The soluble protein content was determined using the Coomath bright blue method [31]. Free amino acids were determined through the ninhydrin method [41]. The lycopene content was determined using the GB/T 14215-2008 method [42]. Vitamin C (VC) was determined using the molybdenum blue colorimetric method [43].

### 2.4. Statistical Analysis

All data were analyzed using SPSS software (ver. 22.0) and presented as the mean ± SD. The data were analyzed through ANOVA, and the mean differences were evaluated by Duncan's multiple range test ($p < 0.05$).

## 3. Results

### 3.1. Effects of Exogenous Melatonin on Plant Growth under Drought Stress

As shown in Table 2, under drought stress, tomato seedlings' plant height, stem diameter, and biomass all drastically declined; compared with the CK treatment, the plant height, stem diameter, and fresh and dry weights of aboveground and underground parts were reduced by 24.32%, 19.02%, 50.61%, 20.27%, 40.35%, and 13.35%, respectively. The treatment of exogenous melatonin alleviated the negative effects of drought to some

extent. Compared with the CK treatment, the T1 treatment increased the plant height, stem diameter, and fresh and dry weights of aboveground and underground parts by 12.88%, 5.98%, 12.29%, 15.55%, 10.79%, and 12.17%. Under drought stress, compared with the T2 treatment, tomato seedlings' plant height, stem diameter, fresh and dry weights of the shoot, and subterranean fresh weight rose by 16.67%, 7.39%, 13.31%, 10.58%, and 28.26%, respectively; the difference in underground dry weight between the two treatments was not significant (Table 2). Comparing the growth of the plants, the MT treatment promoted seedling growth (Figure 1).

**Table 2.** Effects of exogenous melatonin on the growth of tomato plants under drought stress. The values are mean $\pm$ SE (n = 5). Different letters are significant differences at $p < 0.05$ level (Duncan's multiple range test).

| Treatment | Height (cm) | Stem Diameter (mm) | Aboveground Fresh Weight (g) | Aboveground Dry Weight (g) | Underground Fresh Weight (g) | Underground Dry Weight (g) | Relative Water Content of Leaves |
|---|---|---|---|---|---|---|---|
| CK | 11.10 $\pm$ 0.21 [b] | 3.68 $\pm$ 0.16 [b] | 3.2439 $\pm$ 0.136 [b] | 0.2289 $\pm$ 0.015 [b] | 0.5772 $\pm$ 0.040 [b] | 0.0337 $\pm$ 0.002 [b] | 0.8475 $\pm$ 0.016 [a] |
| T1 | 12.53 $\pm$ 0.92 [a] | 3.90 $\pm$ 0.21 [a] | 3.6426 $\pm$ 0.209 [a] | 0.2645 $\pm$ 0.018 [a] | 0.6395 $\pm$ 0.046 [a] | 0.0378 $\pm$ 0.003 [a] | 0.8459 $\pm$ 0.018 [a] |
| T2 | 8.40 $\pm$ 0.64 [d] | 2.98 $\pm$ 0.05 [d] | 1.6023 $\pm$ 0.162 [d] | 0.1825 $\pm$ 0.009 [d] | 0.3443 $\pm$ 0.037 [d] | 0.0292 $\pm$ 0.002 [c] | 0.6637 $\pm$ 0.051 [c] |
| T3 | 9.80 $\pm$ 0.65 [c] | 3.20 $\pm$ 0.12 [c] | 1.8155 $\pm$ 0.077 [c] | 0.2018 $\pm$ 0.013 [c] | 0.4416 $\pm$ 0.056 [c] | 0.0309 $\pm$ 0.004 [b,c] | 0.7520 $\pm$ 0.023 [b] |

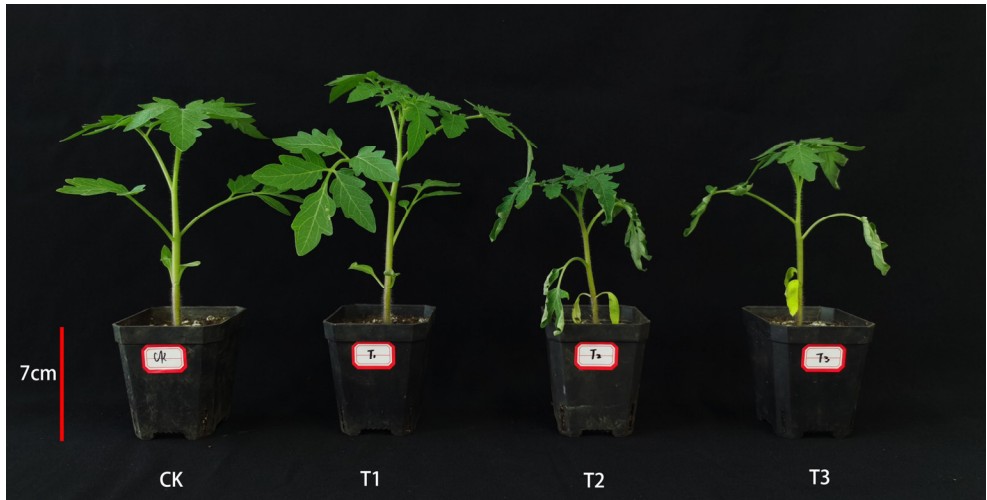

**Figure 1.** Effects of exogenous melatonin on growth state of tomato seedlings under drought stress. CK, control; T1, normal watering + 100 µmol/L melatonin; T2, drought; T3, drought + 100 µmol/L melatonin.

In comparison to CK, the relative water content of T2 decreased by 21.69% under drought stress. Under normal watering conditions, melatonin had little influence on seedlings' relative water content, and there was no significant difference between the CK and T1 treatments. Under drought stress, the water content of seedlings treated with T3 increased by 13.31% compared with that of seedlings treated with T2. Melatonin has the potential to effectively reduce the effect of drought on seedling water content (Table 2).

### 3.2. Effects of Exogenous Melatonin on Chlorophyll Content and Photosynthesis of Tomato Seedlings under Drought Stress

In Figure 2A, drought stress reduced the chlorophyll content, especially when compared to the CK treatment, as the chlorophyll content of T2 decreased by 55.41%. After the seedlings were treated with melatonin, under normal water conditions, the chlorophyll content of the T1 treatment was not statistically different from the CK treatment, but it considerably increased in drought conditions, while the T3 treatment's chlorophyll content was 40.51% greater than the T2 treatment (Figure 2A).

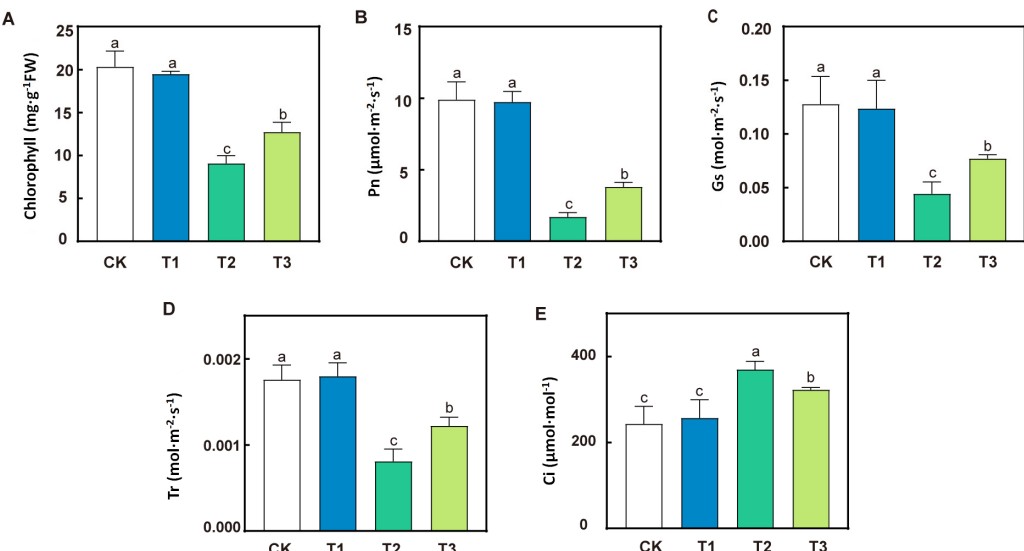

**Figure 2.** Effect of exogenous melatonin on chlorophyll content and photosynthesis of tomato seedlings under drought stress. (**A**) Chlorophyll; (**B**) Net photosynthetic rate (Pn); (**C**) Stomatal conductance (Gs); (**D**) Transpiration rate (Tr); (**E**) Intercellular $CO_2$ concentration (Ci). The values are mean $\pm$ SE (n = 5). Different letters are significant differences at *p* < 0.05 level (Duncan's multiple range test).

Drought stress considerably reduced the Pn, Gs, and Tr of tomato seedlings by 82.75%, 65.42%, and 53.95%, respectively, and the Ci increased by 51.71% (Figure 2B–E). Melatonin alleviated the inhibition of drought stress on photosynthesis, and Pn was 1.2 times higher than that of the T2 treatment. The Gs and Tr increased by 74.11% and 50.86%, respectively, and the Ci decreased by 12.63% (Figure 2B–E).

*3.3. Effects of Exogenous Melatonin on Relative Conductivity, MDA, and Reactive Oxygen Species of Tomato Seedlings under Drought Stress*

As mentioned in Figure 3A,B, under drought stress (T2), there was a significant accumulation of active oxygen, leading to an increase in the contents of $H_2O_2$ and $O_2^-$ by 81.21% and 68.43% when compared to the control group (CK). The DAB staining diagram (Figure 3E) and NBT staining diagram (Figure 3F) also demonstrate that drought stress promoted the accumulation of $H_2O_2$ and $O_2^-$ active oxygen. Melatonin treatment did not affect the content of $O_2^-$ in tomato seedlings grown in normal water, but it did increase the level of $H_2O_2$. When comparing the T3 and T4 groups, it can be observed that melatonin simultaneously reduced the level of $H_2O_2$ and $O_2^-$ by 17.28% and 16.11%, respectively (Figure 3A,B), under drought stress. These findings were further confirmed by the DAB (Figure 3E) and NBT (Figure 3F) staining images. Overall, the results suggest that under drought stress conditions, melatonin treatment effectively mitigated the accumulation of active oxygen, reducing the levels of both $H_2O_2$ and $O_2^-$ in tomato seedlings.

Drought substantially increased the relative electrical conductivity and MDA content of seedlings (Figure 3C,D). In comparison to the CK treatment, the relative electrical conductivity and MDA content of seedlings under T2 treatment increased by 64.16% and 43.14%, respectively. In normal water conditions, melatonin treatment had no significant effect on the relative conductivity or MDA content, while the T1 treatment and CK treatment made no significant difference. The T3 treatment considerably reduced the content of each substance by 20.90% and 21.44% when compared to the T2 treatment.

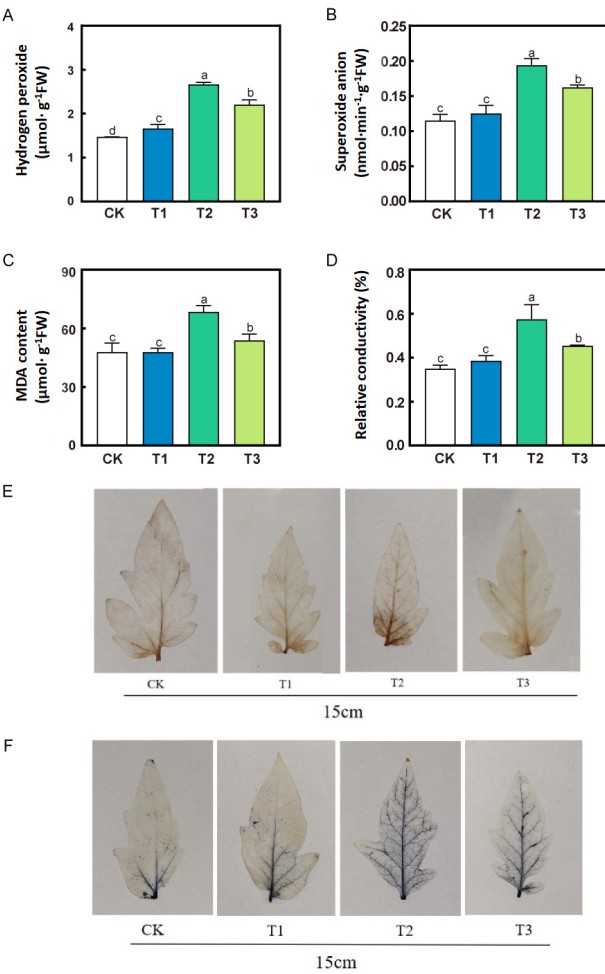

**Figure 3.** Effect of exogenous melatonin on (**A**) hydrogen peroxide contents, (**B**) superoxide anion contents, (**C**) malondialdehyde, (**D**) relative conductivity, (**E**) $H_2O_2$ by 3, 3-diaminobenzidine (DAB), and (**F**) histochemical analysis for $O_2^-$ by nitro blue tetrazolium (NBT) of tomato seedlings under drought stress. The values are mean $\pm$ SE (n = 5). Different letters are significant differences at $p < 0.05$ level (Duncan's multiple range test).

### 3.4. Effects of Exogenous Melatonin on the Activities of Osmotic Regulators and Antioxidant Enzymes under Drought Stress

The antioxidant enzyme activity of tomato seedlings rose extensively under drought stress; the SOD and POD increased by 35.08% and 10.26%, respectively, and the CAT increased 1.92 times (Figure 4A–C). Under normal water conditions, melatonin had no influence on antioxidant enzyme activity, while under stress conditions, the activity of antioxidant enzymes increased significantly after the application of melatonin. Compared with the T2 treatment, the SOD and POD in the T3 treatment increased by 34.74% and 35.71%, respectively (Figure 4A,B). Despite the fact that there was no statistically significant difference in the CAT content change, it increased by 60.52% compared with the T2 treatment (Figure 4C).

As evident from Figure 4D–F, drought stress significantly increased the proline content, soluble protein, and soluble sugar content of seedlings. The proline content of the T2 treatment increased 11.13 times, the soluble protein content increased by 60.07%, and the soluble sugar level increased 1.19 times compared with the CK treatment. Melatonin had no influence on proline, soluble sugar, or soluble protein levels under normal water conditions, and there was little difference between the T1 and CK treatments. The T3 treatment significantly reduced the content of various substances compared with the T2 treatment, with a decrease of 27.82%, 30.30%, and 23.24% under drought stress (Figure 4D–F).

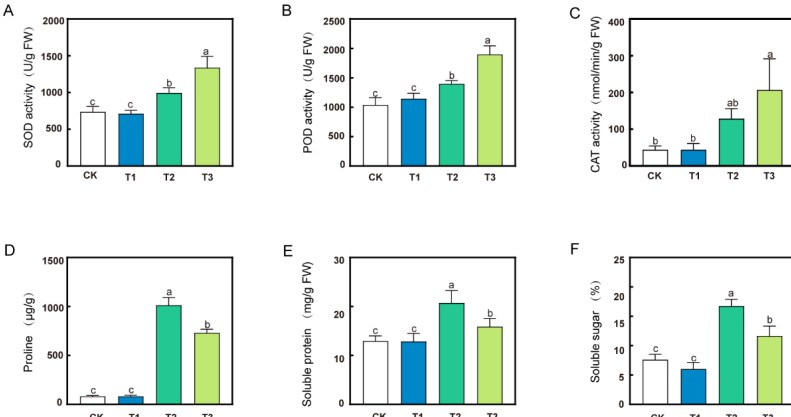

**Figure 4.** Effect of exogenous melatonin on (**A**) SOD, (**B**) POD, (**C**) CAT, (**D**) proline, (**E**) soluble protein, and (**F**) soluble sugar of tomato seedlings under drought stress. The values are mean $\pm$ SE (n = 5). Different letters are significant differences at $p < 0.05$ level (Duncan's multiple range test).

### 3.5. Effects of Exogenous Melatonin on ABA Content and Related Synthetic Genes under Drought Stress

Under drought stress, the content of ABA in tomato seedling leaves and roots was enhanced considerably—4.3 times and 16.4 times, respectively (Figure 5A,B). Furthermore, the expression of the ABA synthesis genes NCED1, NCED2, and NCED3 rose significantly. Compared with the control, the expression of the NCED1 gene in the leaves increased about 4–6 times, the expression of the NCED1 gene in the roots increase approximately 2.5 times, and the expression of NCED2 and NCED3 increased about 7 times (Figure 5C,D). After melatonin was applied to the roots under normal water conditions, the ABA content in the leaves compared to the control group did not differ significantly, and, with the exception of NCED3, there was no discernible variation in the expression of NCED1 and NCED2. No discernible change existed despite the roots' greater expression of NCED1 and NCED2, as well as their increased ABA content, when compared to the control. Under drought stress, compared with the T2 treatment, the content of ABA in the leaves and roots of the T3 treatment was significantly reduced by 28.38% and 9.88%, respectively, and each gene's expression was markedly decreased compared to the T2 treatment (Figure 5A–D).

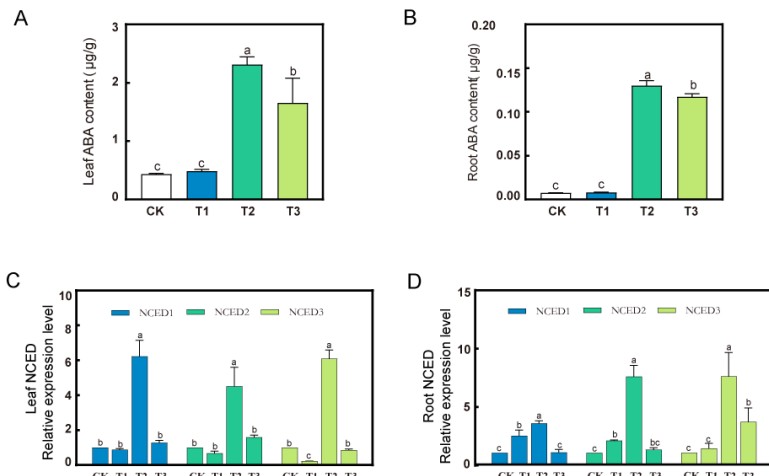

**Figure 5.** Effect of exogenous melatonin on ABA content and the expression of related genes in tomato seedlings under drought stress. (**A**,**B**) leaf ABA content and root ABA content. (**C**,**D**) The relative expression level of leaf three ABA-synthesis-related genes NCED1, NCED2, and NCED3 (nine-cis-epoxycarotenoid dioxygenase) in root and leaf. ABA content and root ABA content The values are mean $\pm$ SE (n = 3). Different letters are significant differences at $p < 0.05$ level (Duncan's multiple range test).

### 3.6. Effects of Exogenous Melatonin on Root of Tomato Seedlings under Drought Stress

Drought stress reduced the root activity of tomato seedlings, and when compared to CK treatment, root activity under T2 treatment dropped by 56.9% (Figure 6A). After the seedlings were treated with melatonin, the root activity of T1 treatment increased by 31.88% compared with CK treatment under normal water conditions, root activity of T3 treatment grew by 52.21% in comparison to T2 treatment under drought conditions (Figure 6A).

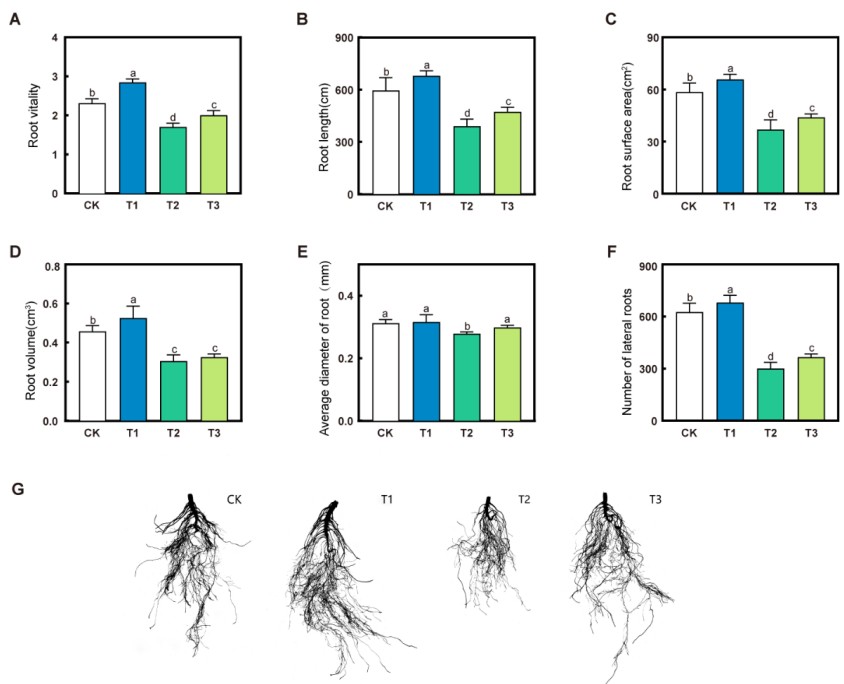

**Figure 6.** Effects of exogenous melatonin on root activity and root morphology of tomato seedlings under drought stress. (**A**) Root vitality; (**B**) root length; (**C**) root surface area; (**D**) root volume; (**E**) average diameter of root; (**F**) number of lateral roots; (**G**) root phenotype. The values are mean ± SE (n = 5). Different letters are significant differences at $p < 0.05$ level (Duncan's multiple range test).

Drought stress significantly reduced the total root length, root surface area, root volume, average root diameter, and lateral root number. Compared with CK, the root length, root surface area, average root diameter, root volume, and lateral root number of the T2 treatment decreased by 34.62%, 36.85%, 10.89%, 33.37%, and 52.10%, respectively, while the average root length increased significantly, by 29.75% (Figure 6B–F). Melatonin treatment can promote the root development of tomato seedlings under both normal watering conditions and drought stress conditions. Melatonin treatment significantly increased the root length and root surface area of seedlings. In comparison to CK, the root length and root surface area of T1-treated seedlings was raised by 14.25% and 12.44%, respectively, and that of T3-treated seedlings increased by 21.43% and 19.03%, respectively (Figure 6B,C). Under drought stress, melatonin increased the average root diameter of seedlings, and the average root diameter of the T3 treatment increased by 7.19% compared with that of the T2 treatment; nevertheless, there was no discernible difference between T1 and CK under normal water distribution conditions (Figure 6E). Compared with the CK treatment, the root volume of the T1 treatment increased by 14.86%, while there was no discernible difference between the root volume of the T3 and the T2 treatments (Figure 6D). Compared with CK, the number of lateral roots of seedlings treated with T1 increased by 8.84%, and the number of lateral roots of the T3 treatment was significantly higher than that of the T2 treatment, with an increase of 22.18% (Figure 6F). It is clear that melatonin dramatically promoted root development (Figure 6G).

### 3.7. Effects of Exogenous Melatonin on Tomato Yield and Single Fruit Weight under Drought Stress

Drought stress significantly reduced tomato fruit yield and single fruit weight by 78.19% and 52.60%, respectively. The application of melatonin increased the fruit yield and single fruit weight under normal water conditions and water deficit conditions by 45.24%, 34.07%, 61.92%, and 37.79%, respectively (Figure 7A,B).

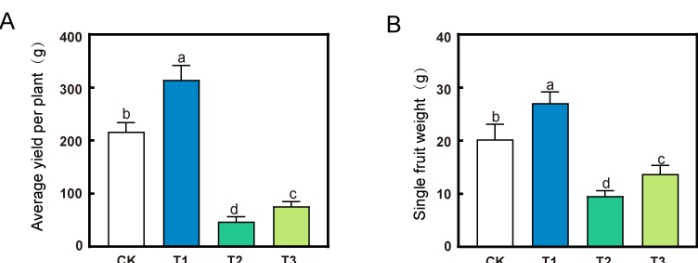

**Figure 7.** Effects of melatonin on (**A**) average yield per plant and (**B**) single fruit weight of tomato under drought stress. The values are mean ± SE (n = 5 for average yield per plant, n = 10 for single fruit weight). Different letters are significant differences at *p* < 0.05 level (Duncan's multiple range test).

### 3.8. Effect of Exogenous Melatonin on Tomato Fruit Quality under Drought Stress

As shown in Figure 8A, drought stress increased tomato fruit quality to a certain extent, and melatonin further increased fruit quality. Under drought stress, the contents of soluble solid, soluble sugar, and organic acid increased in comparison to CK by 43.75%, 32.68%, and 83.33%, respectively. Under normal water conditions, melatonin increased the soluble solids by 30%, while the soluble sugar and organic acid content did not change significantly. After applying melatonin after drought stress, the T3 treatment significantly reduced the organic acid by 23.77% compared with the T2 treatment, and the soluble solid and soluble sugar increased by 15.07% and 35.49%, respectively (Figure 8A).

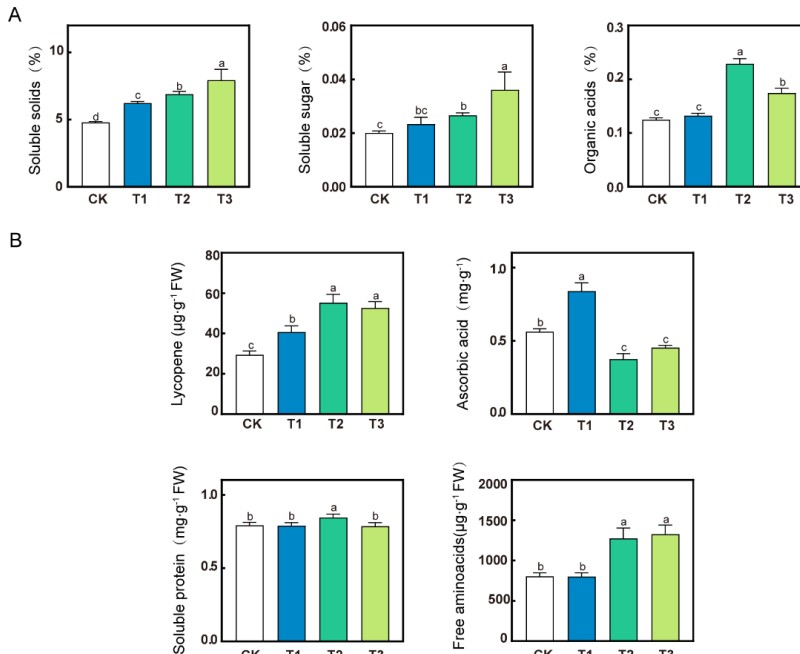

**Figure 8.** Effects of exogenous melatonin on tomato fruit quality under drought stress. (**A**) Soluble solid, soluble sugar, and organic acid. (**B**) Lycopene, ascorbic acid, soluble protein, and free amino acid. The values are mean ± SE (n = 5). Different letters are significant differences at *p* < 0.05 level (Duncan's multiple range test).

Under drought stress, the content of lycopene, soluble protein, and free proline increased by 87.48%, 6.80%, and 58.14%, respectively, and the content of vitamin C decreased by 33.27% compared with the control (Figure 8B). Melatonin can promote fruit quality under normal watering conditions. Compared with the CK treatment, vitamin C and lycopene in the T1 treatment increased by 49.23% and 38.09%, respectively. After applying melatonin under drought conditions, the T3 treatment significantly reduced the soluble protein content by 7.10% compared with the T2 treatment, while the content of vitamin C, lycopene, and free amino acid did not change significantly (Figure 8B).

## 4. Discussion

Under drought stress, the growth and development of tomato were affected. In this study, the plant height, stem diameter, and biomass of tomato seedlings were dramatically reduced under drought stress. On the contrary, melatonin reduced the growth inhibition of tomato seedlings and increased the plant height, stem diameter, and biomass, which is consistent with previous studies [44]. This study's use of exogenous melatonin also increased the water content of plants, significantly alleviated the decline of relative water content in leaves, and increased water use efficiency.

### 4.1. Exogenous Melatonin Alleviated the Inhibition of Photosynthesis by Drought Stress and Increased the Yield of Tomato

In plant photosynthesis, chlorophyll is crucial to the transfer, distribution, and transformation of light energy. The photosynthetic rate of photosynthetic organisms and the rate of organic matter accumulation are closely related to the chlorophyll content in photosynthetic tissues [45]. Chlorophyll content can be dramatically impacted by drought stress. Some researchers believe that maintaining high chlorophyll content is one of the measures to improve crop yield [46]. According to the findings, melatonin increased the chlorophyll content of tomato seedlings under drought stress, but under normal conditions, there was no discernible difference.

Under drought stress, plant photosynthetic performance was inhibited. Plants exchange water and gas with the environment mainly through stomata, and stomatal resistance increases under drought, which leads to stomatal closure, resulting in adverse effects on photosynthesis and transpiration [47]. Plants can reduce water consumption by reducing Gs under drought stress. Gs and Ci displayed opposing patterns; this is because the structure and function of photosynthetic organs are destroyed, and the fixation of intercellular $CO_2$ is decreased under stress [48]. In this research, drought stress significantly decreased the Pn, Gs, and Tr and increased the Ci level, while melatonin increased the Pn, Gs, and Tr; the findings suggested that through enhancing stomatal features, melatonin could enhance photosynthesis, transpiration, stomatal conductance, and water use efficiency.

### 4.2. Exogenous Melatonin Reduced the Damage of Plasma Membrane of Seedlings under Drought Stress

Plant cells may sustain oxidative damage from an excessive accumulation of reactive oxygen species. Superoxide anion ($O_2{}^-$) and hydrogen peroxide ($H_2O_2$) are two species of reactive oxygen species produced during plant metabolism. Under drought stress, tomato seedlings quickly developed a lot of reactive oxygen species, and the accumulation of reactive oxygen species was intensified, the degree of membrane lipid peroxidation was increased, membrane permeability was changed, cell physiological function was disturbed, and, finally, cell death was caused [49]. Relative conductivity and MDA can be utilized to assess the degree of membrane peroxidation and damage [50]. In this study, melatonin reduced reactive oxygen species, relative conductivity, and malondialdehyde content under drought stress, indicating that melatonin alleviated membrane damage in seedlings.

Also, many studies have shown that melatonin is an active antioxidant involved in the regulation of abiotic and biotic stress tolerance in horticultural crops. Due to its lipophilic and hydrophilic properties, melatonin can play an antioxidant role in cells and even in the nucleus [51]. Under drought stress, exogenous melatonin can activate the

antioxidant system of apple, removing excess reactive oxygen species (ROS), improving the efficiency of light energy conversion and carbon assimilation in leaves, and then alleviating drought symptoms [52]. It reduced lipid peroxidation and ROS accumulation in Moringa oleifera [53], promoted the antioxidant enzymes system in fenugreek [54], increased glutathione and ascorbic acid content in kiwifruit seedlings [55], enhanced osmoregulatory substances in tomato [19], protected the grana lamella of chloroplast in wheat [56], preserved the chloroplast structure in Brassica napus [57], and significantly regulated the antioxidant enzymes (APX, CAT, DHAR, GST, GR, MDHAR, POD, SOD) and non-enzymatic antioxidant (AsA, GSH) genes in Solanum lycopersicum and Carya cathayensis [21,58]. Under drought stress, a large amount of proline, soluble protein, and soluble sugar were accumulated in seedlings, and the contents of all substances were decreased to varying degrees after melatonin application. Some researchers believe that proline is a sign of stress damage, and because proline synthesis requires a lot of energy, more energy can be saved to cope with stress by reducing proline content [59]. The experimental results showed that melatonin could reduce the degree of membrane damage caused by drought, improve the resistance of seedlings, and promote the normal growth of seedlings by regulating the content of osmotic regulatory substances and the activity of antioxidant enzymes.

*4.3. Exogenous Melatonin Enhanced Drought Resistance by Reducing Endogenous ABA Content*

Plant stress resistance is greatly influenced by plant hormones, and research has indicated that exogenous melatonin can regulate changes in plant hormone content and enhance plant stress resistance. ABA is one of the main endogenous hormones in plants, which has the function of regulating plant development, inhibiting seed germination, and promoting leaf senescence. With the advancement of research, ABA was discovered to be crucial in the development of the plant's stress response. In this study, drought stress caused the ABA content in leaves and roots to increase several times, indicating that ABA participated in the response process of tomato seedlings to drought stress. Studies have shown that ABA has a dual effect on plant growth and development, promoting plant senescence and leaf shedding at high concentrations and promoting growth and stress resistance at low concentrations [60]. Melatonin significantly decreased ABA content under drought stress, and the drop in ABA content was associated with the down-regulation of ABA synthesis gene expression. The up-regulation of NCED1, NCED2, and NCED3 gene expression was induced by drought, and the down-regulation of NCED1, NCED2, and NCED3 gene expression was induced by melatonin. Therefore, melatonin can slow down the senescence process of tomato seedlings by lowering ABA content and down-regulating the expression of ABA synthesis genes, which is consistent with previous studies. Melatonin decreased ABA content and ABA biosynthesis and down-regulated signaling pathway factors under high-temperature stress [61]. Low concentrations of ABA cause stomatal closure to reduce water loss, while also controlling the expression of genes linked to dehydration tolerance to keep water levels normal [62].

*4.4. Exogenous Melatonin Promoted Plant Growth under Drought Stress by Regulating Root Configuration, thus Promoting the Formation of Yield and Quality*

Under drought conditions, the root system acts first in response to stress. When the root system senses the stress signal, the expression of stress-related enzymes and proteins in the plant changes, having an impact on the distribution ratio of carbon assimilation products in various organs and, finally, having an impact on root growth [63]. Drought stress can seriously affect plant roots, root biomass, total root length, and root volume, and the average root diameter of soybean was drastically decreased under drought conditions [64]. According to studies, melatonin is a significant regulatory factor controlling root growth, and it has a regulatory influence on root growth [13,24,65].

Although auxin is the main hormone regulating root growth, ABA has been shown to play a crucial part in controlling root growth. ABA can regulate root growth by affecting

the quantity and size of apical dividing cells [66,67]. Meanwhile, ABA plays a key role in plant root growth and development. Endogenous ABA biosynthesis can significantly inhibit lateral root formation [68]. Duan et al. found that lateral roots have a stronger inhibitory effect than taproots under salt stress. This is due to differences in ABA signal transduction [69]. In this study, the application of melatonin significantly increased the root vitality of tomato seedlings, and, at the same time, root morphological parameters, such as total root length, root surface area, mean root diameter, and the number of lateral roots, were significantly increased and the root morphology was improved. Melatonin dramatically decreased ABA content and suppressed the expression of synthetic genes under drought, suggesting that melatonin might regulate root growth by mediating ABA content and its anabolism level and alleviate the adverse effects of drought on lateral root development.

Root system enhancement regulates the ability of plants to attain nutrients and water. Root parameters directly affect plants' capacity to absorb and transport available water and nutrients [70]. Root configuration and expansion mainly depend on the growth of lateral roots, which is conducive to increasing the contact area with soil, thus improving the water and nutrient capture, absorption, and utilization efficiency of plants. Photosynthesis is the most fundamental physiological function of plants, and improving the photosynthetic function of crops is a good basis for improving crop yield [71]. However, the photosynthetic function is affected by many aspects, among which water is the raw material of plant photosynthesis, and mineral elements directly or indirectly affect the photosynthesis to some extent. Under drought conditions, melatonin enhanced the number of lateral roots, improved the root morphology, and supported the obtention of additional moisture and nutrients from the soil. The seedlings had sufficient nutrients and robust growth, which further increased the photosynthetic characteristics of tomato and increased the output of photosynthetic products, which was conducive to the improvement of yield and quality.

## 5. Conclusions

In this study, we have demonstrated that the application of exogenous melatonin can enhance the resistance of tomato seedlings to drought stress. This is achieved by regulating the content of osmotic regulators and the activity of antioxidant enzymes, which ultimately leads to improved growth and development of plants under drought stress. Additionally, as far as we know, this study is the first to show that melatonin also plays a role in regulating root growth through mediating endogenous ABA biosynthesis and metabolism. This leads to improved water and nutrient absorption and utilization efficiency in tomato plants, resulting in enhanced fruit yield and quality under drought stress. However, further research is needed to investigate the specific physiological and molecular mechanisms through which melatonin interacts with other hormones to regulate plant drought stress tolerance. Molecular approaches should also be explored in order to gain a more comprehensive understanding.

**Author Contributions:** Q.H., M.Y., J.D. and X.L. designed the experiments and conducted the research; H.Y., M.C. and Y.X. performed the experiments about fruit quality and yield; Q.H., M.Y. and H.Y. performed most of the experiments and analyzed the data; Q.H., H.Y. and X.L. wrote the article. X.H. provided recommendations for experiment conduct; All authors have read and agreed to the published version of the manuscript.

**Funding:** This research was funded by the National Key Research and Development Program of China (2019YFD1000300).

**Data Availability Statement:** Not applicable.

**Acknowledgments:** We thank Juan Xie (Horticulture Science Research Center at College of Horticulture, Northwest A&F University, Yangling, China) for her technical support in leaf photosynthetic data collection.

**Conflicts of Interest:** The authors declare no conflict of interest.

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
