# Peer review of "Enhancing Drought Tolerance and Fruit Characteristics in Tomato through Exogenous Melatonin Application"

_horticulturae, doi:10.3390/horticulturae9101083_

Round 1
Reviewer 1 Report
The manuscript titled “Exogenous melatonin improves drought tolerance and fruit quality and yield in tomato” contains important and interesting information. The application of melatonin effectively promotes the growth and development of tomato seedlings under drought stress. Melatonin improves fruit quality while reducing the effects of drought on yield. In addition, melatonin has been shown to regulate the concentration of ABA and its associated genes.
The abstract clearly and accurately described the content of the article, the results were clearly presented, and the interpretations and conclusions were justified by the results. However, this MS needs to be carefully reviewed and revised before publication in Horticulturea is considered.
Comments are as follows;
- Although the experimental design is suitable to test the hypothesis, it is somewhat difficult to understand, such as from L98-L100, the experiment is divided into two parts... one is the study on the effect of melatonin on drought tolerance of tomato seedlings and the second is the field study to determine the effects of melatonin on fruit yield and quality…Is my understanding correct?
- Line 114- The temperature at which the sample is stored should be indicated
- Line 120-121- At the seedling growth stage, the relative water content of the third leaf was determined…how is it determined? Briefly explain the method used.
- Line 128- The relative conductivity was determined using the immersion method…how?
- Figure 1- Revise the label on the photo of the seedling, make it clearer.
- Table 2- Column 1 ---Treatment
- Line 240-242- under drought stress, melatonin simultaneously decreased H2O2 and O2 content by 17.28% and 16.11%, respectively (Figure 3A and B). The results were also verified by DAB (Figure 3E) and NBT (Figure 3F) staining images...the authors should explain in more detail how they were able to verify this using the staining images?
- References: Please check that all references are correct.
- In this paper, all text should be carefully checked.
As shows in comments and suggestions for authors
Reviewer 2 Report
please open the attached file.

English Language is accepted
Reviewer 3 Report
Review on “Exogenous melatonin improves drought tolerance and fruit quality and yield in tomato” for IJMS (manuscript ID horticulturae-2578879)
In this manuscript the authors present an evaluation of effect of melatonin treatment to the growing tomato during drought stress.
Comments on Introduction section:
Large part of the Intro (L41-62) devoted to the drought stress response of tomato plant, but lacking knowledge to date of tomatoes’ melatonin treatment or previous studies review on the topic. I strongly suggest to include the recent data about the drought impact to the tomato plant and add the review of previous known experiments (melatonin treatment).
The following papers may help to improve the Intro:
· Mushtaq N, Iqbal S, Hayat F, Raziq A, Ayaz A, Zaman W. Melatonin in Micro-Tom Tomato: Improved Drought Tolerance via the Regulation of the Photosynthetic Apparatus, Membrane Stability, Osmoprotectants, and Root System. Life. 2022; 12(11):1922. https://doi.org/10.3390/life12111922
· Ibrahim MFM, Elbar OHA, Farag R, Hikal M, El-Kelish A, El-Yazied AA, Alkahtani J, El-Gawad HGA. Melatonin Counteracts Drought Induced Oxidative Damage and Stimulates Growth, Productivity and Fruit Quality Properties of Tomato Plants. Plants. 2020; 9(10):1276. https://doi.org/10.3390/plants9101276
· Yang L, Bu S, Zhao S, Wang N, Xiao J, He F, et al. (2022) Transcriptome and physiological analysis of increase in drought stress tolerance by melatonin in tomato. PLoS ONE 17(5): e0267594. https://doi.org/10.1371/journal.pone.0267594
· Altaf MA, Shahid R, Ren M-X, Naz S, Altaf MM, Khan LU, Tiwari RK, Lal MK, Shahid MA, Kumar R, et al. Melatonin Improves Drought Stress Tolerance of Tomato by Modulating Plant Growth, Root Architecture, Photosynthesis, and Antioxidant Defense System. Antioxidants. 2022; 11(2):309. https://doi.org/10.3390/antiox11020309
· Annadurai MKK, Alagarsamy S, Karuppasami KM, Ramakrishnan S, Subramanian M, Venugopal PRB, Muthurajan R, Vellingiri G, Dhashnamurthi V, Veerasamy R, et al. Melatonin Decreases Negative Effects of Combined Drought and High Temperature Stresses through Enhanced Antioxidant Defense System in Tomato Leaves. Horticulturae. 2023; 9(6):673. https://doi.org/10.3390/horticulturae9060673
· Xie Q, Zhang Y, Cheng Y, Tian Y, Luo J, Hu Z, Chen G. The role of melatonin in tomato stress response, growth and development. Plant Cell Rep. 2022 Aug;41(8):1631-1650. doi: https://doi.org/10.1007/s00299-022-02876-9
· Jensen NB, Ottosen C-O, Zhou R. Exogenous Melatonin Alters Stomatal Regulation in Tomato Seedlings Subjected to Combined Heat and Drought Stress through Mechanisms Distinct from ABA Signaling. Plants. 2023; 12(5):1156. https://doi.org/10.3390/plants12051156
· Kaya, C., Ugurlar, F., Ashraf, M., Alyemeni, M. N., & Ahmad, P. (2023). Exploring the synergistic effects of melatonin and salicylic acid in enhancing drought stress tolerance in tomato plants through fine-tuning oxidative-nitrosative processes and methylglyoxal metabolism. Scientia Horticulturae, 321, 112368. https://doi.org/10.1016/j.scienta.2023.112368
The refences in Intro section should be replaced with those which related to tomato plants or closer species.
My questions about Results and Discussion:
Unfortunately, in the Discussion section authors didn’t compare their results with other studies on the topic. Melatonin treatment is quite popular practice during drought stress. Please review the studies on the topic and extend the Discussion with the comparison.
L367-371: this fragment better fits for the Intro section
L376: references [37, 38] are not related with tomato plants.
L413: please add/replace the reference here, [46] devoted to humans (!)
Conclusion section should contain the summary of the findings, not the speculation for the further research (L571-L584).
Methods section comments:
· L90: please describe the irrigation regime for CK and drought experiments.
· L92-94: please avoid repetition (“treatment”)
· L123: is the 1949-year paper [21] describes the same technique of chlorophyll content measurement that authors used? I found more recent methods paper https://doi.org/10.1080/02757259009532129
· L133: reference [27] doesn’t cover hydrogen peroxide determination.
· L167: why “actin” gene was selected as reference gene? Please explain
Some minor corrections to the text (style and spelling):
· L130: “contentt” → “content”.
· L147, 154: substances names start with capital letters
· L437: remove extra comma
· L182: explain what “VC” means
· There are some irrelevant numbers and color highlights in the Reference section (page 16)

Reviewer 4 Report
The study conducted by Huang et al., titled "Exogenous melatonin improves drought tolerance and fruit quality and yield in tomato", holds significant importance as it delves into Melatonin, a pivotal compound in stress tolerance. The research aimed to evaluate Melatonin's impact on aspects like growth, root structure, abscisic acid (ABA) content, yield, and fruit quality during drought stress. The outcomes yielded great promise, revealing Melatonin's capacity to enhance seedling growth, elevate photosynthesis, augment chlorophyll levels, diminish reactive oxygen species and osmotic regulators, amplify antioxidant enzyme activity, and bolster adaptability to drought stress. Among the discussed mechanisms tied to Melatonin treatment, one crucial facet is its direct influence on reducing ABA levels. Furthermore, other mechanisms encompass the reduction of ABA-linked gene expression, indicating its role in governing root growth through mediation of ABA biosynthesis. This intervention heightened root vitality, spurred growth, and ultimately elevated both tomato fruit yield and quality.
The paper fits perfectly into the scope of the journal, and is well-structured and well-written. some minor remarks must be taken into consideration by the authors before considering this work for publication.
1. Title : I suggest the following title "Enhancing Drought Tolerance and Fruit Characteristics in Tomato through Exogenous Melatonin Application", the revised title succinctly conveys the main objective of the study. It explicitly highlights the core focus: the augmentation of both drought tolerance and fruit characteristics in tomato plants. This clarity provides readers with an immediate understanding of what the research seeks to accomplish.
2. Introduction : Please rectify the family name to Solanaceae family.
The objectives of the study are mentioned at the end, which is great. However, you might want to reiterate more explicitly that the purpose of the study is to investigate the effects of exogenous melatonin on tomato seedlings' growth, root morphology, endogenous ABA content, and yield and quality under drought stress.
3. Materials and Methods :
subsection 2.1. Experimental materials
* The process of "cleaning, disinfection, seed soaking, germination" should be elaborated upon. The duration, methods, and purpose of each step need to be provided. This is vital to ensure that readers can replicate the study if necessary.
* The term "two leaves and one heart" is not universally understood. You should provide a clearer description of the growth stage
Subsection 2.2. Experimental designs
The methodology and experimental process are explained in sufficient detail. However, it lacks details on how was the melatonin solution prepared?, and how was the irrigation carried out?
The comprehensive analysis and interpretation of the results are highly commendable. The thorough examination of the outcomes provides valuable insights into the study's findings. The authors should rephrase the conclusion section to align it with the acquired results and offer some future prospects stemming from the study.
Round 2
Reviewer 2 Report
The authors did all the correction and respond well in all comments. Therefore i recommending to publish the manuscript in Horticulturae journal.
the English Languag is accepted
Author Response
Dear reviewer:
Thank you very much for your careful guidance. We have involved a native English speaker to retouch the manuscript. Thanks again for your time and kind consideration.
Reviewer 3 Report
Review on “Exogenous melatonin improves drought tolerance and fruit quality and yield in tomato” for IJMS (manuscript ID horticulturae-2578879)
I would to thank authors for the efforts to improve the manuscript, but it still requires some corrections.
I suggest to revise the keywords – include drought and tomato (Solanum lycopersicum).
Methods section comments:
· L109: the irrigation regime for T2 and T3 is still unclear, please specify the severity of the drought.
·
Some minor corrections to the text (style and spelling):
· L96: “Sodium” → “sodium”.
· L153: please explain the acronym “FWS”
· L173: “]” – missing reference?
· L199: did you mean “Liquid chromatography–mass spectrometer”?
· L220: please specify version and reference to Win RHIZO software
· L230: “Dun-can’s” → “Duncan’s”
· L455, L522: please remove the brackets
· L462: “reduced” → “Reduced”
